# Two New Species of *Mesochorista* (Insecta, Mecoptera, Permochoristidae) from the Guadalupian Yinping Formation of Chaohu, Eastern China [note 1]

**DOI:** 10.3390/insects16111130

**Published:** 2025-11-05

**Authors:** Xinneng Lian, Chenyang Cai, Zhuo Feng, Diying Huang

**Affiliations:** 1Institute of Palaeontology, Yunnan Key Laboratory of Earth System Science, Yunnan University, Kunming 650500, China; xnlian@nigpas.ac.cn (X.L.);; 2State Key Laboratory of Palaeobiology and Stratigraphy, Nanjing Institute of Geology and Palaeontology, Chinese Academy of Sciences, Nanjing 210008, China; 3Southwest United Graduate School, Kunming 650092, China

**Keywords:** new taxa, permochoristids, scorpionflies, Permian, Yangtze platform

## Abstract

The megadiverse Permian mecopteran family Permochoristidae is well recorded worldwide and plays an important role in understanding the origin of the crown group of Mecoptera. However, fossil records from China remain scarce. Herein, we describe and illustrate two new species of *Mesochorista* from the Guadalupian Yinping Formation: *Mesochorista tillyardi* Lian and Huang, sp. nov., and *Mesochorista yinpingensis* Lian and Huang, sp. nov. In addition, we discuss *Permochorista* and *Mesochorista* and support that *Permochorista* should be considered a junior synonym of *Mesochorista*. These new fossils extend the known distribution of *Mesochorista* and improve our understanding of its diversity.

## 1. Introduction

Mecoptera are one of the oldest holometabolous insect orders that can be dated back to the early Permian [1]. Recent Mecoptera exist as a relict lineage comprising only ca. 800 species [2,3]. In contrast, fossil mecopterans were diverse and constituted a significant component of various Paleozoic and Mesozoic entomofaunas [4,5]. Mecopterans are commonly referred to by different vernacular names based on their morphological, ecological, and ethological traits. For example, species with enlarged and upward-curved male genitalia, resembling a scorpion’s tail, are called “scorpionflies” (e.g., Panorpoidea Latreille, 1805), species that hang from vegetation using their forelegs while capturing prey are known as “hangingflies” (Bittacidae Handlirsch, 1906, Cimbrophlebiidae Willmann, 1977), those with forceps-like male genitalia reminiscent of earwigs are referred to as “earwigflies” (Meropeidae Handlirsch, 1906), and species adapted to cold environments and often found on snow surfaces are termed “snow scorpionflies” (Boreidae Latreille, 1816).

Mecoptera reached their first peak in species diversity during the Middle to Late Permian, with fossil records reported from Africa, Australia, Brazil, China, India, North America, and Russia [6,7,8,9,10,11,12,13,14,15]. The family Permochoristidae Tillyard, 1917 was the dominant mecopteran group during Permian; however, their diversity declined significantly after the onset of the Triassic and only survived as a relict lineage into the Jurassic, with the latest record being from the Early Jurassic [16,17]. The classification systems of Permochoristidae essentially rely on wing venation. The most common character of this diverse family is the six-branched M (M_2_ and M_4_ with a fork), while the other venational characters display a fairly high disparity. Based mainly on the branches of Sc and Rs, Permochoristidae was divided into four subfamilies: Permochoristinae Tillyard, 1917, Agetopanorpinae Carpenter, 1930, Sylvopanorpinae Novokshonov, 1997, and Pseudonannochoristinae Novokshonov, 1994. The Permochoristinae, which represents one of the most diverse mecopteran subfamilies, is characterized by its Sc possessing one or two anterior veinlets and Rs forking into four branches.

Over the past years, we have reported several Permian mecopterans from the Guadalupian Yinping Formation [15,18,19], including two permochoristid species: *Permeca chaohuensis* Lian, Cai and Huang, 2022 (Permochoristinae), and *Chaohuchorista liaoi* Lian, Cai and Huang, 2022 (Pseudonannochoristinae). However, records of Permochoristidae from China are still scarce compared with their high diversity in other deposits worldwide. In this report, two new species of *Mesochorista*, 1916 are described and illustrated from the Yinping Formation, which represent the first record of *Mesochorista* from the Permian of China.

## 2. Materials and Methods

The studied specimens were collected from black shale of the lower part of the Yinping Formation near Houdong Village, Chaohu City, Anhui Province (see detailed map in Lian et al. [15]: Figure 1). The Yinping Formation is considered to be Guadalupian in age (see detailed discussions in Lian et al. [18]).

Some specimens were carefully prepared using a sharp knife. Photographs were captured by a digital camera attached to a Zeiss Discovery V20 microscope, and scanning electron microscopy (SEM) images were obtained with a Hitachi SU 3500 scanning electron microscope, operating with an accelerating voltage of 25 kV and a pressure of 60 Pa. Line drawings were made using Adobe Illustrator 2019 graphic software (San Jose, CA, USA). All specimens are housed in the Nanjing Institute of Geology and Paleontology, Chinese Academy of Sciences, Nanjing, China.

The venation terminology primarily follows the scheme proposed by Minet et al. [20] and partly follows Bashkuev and Sukatsheva [21].

## 3. Systematic Paleontology

Order Mecoptera Packard, 1886

Family Permochoristidae Tillyard, 1917

Subfamily Permochoristinae Tillyard, 1917

Genus *Mesochorista* Tillyard, 1916


** **


**Revised diagnosis.** Forewings ca. 6–11 mm long; Sc with one (Sc_2_) or two anterior veinlets (Sc_2_ and Sc_3_), Sc_3_ usually faint or absent; Rs with four branches, Rs_1+2_ generally more than twice as long as Rs_3+4_; M usually with six branches (sometimes five).

**Remarks.** *Mesochorista* is a derived permochoristid group showing a trend toward venational simplification. It commonly remains a forked M_2_, a plesiomorphic character; while it also shares some derived characters, such as a simplified Sc (Sc_2_ and Sc_3_ shortened and weakened), and Rs (with Rs_4_ unbranched). In some species, M_2_ is single, a condition characteristic of more derived crown-group Mecoptera since the Mesozoic.


** **


*Mesochorista tillyardi* Lian and Huang, sp. nov.

(Figure 1, Figure 2 and Figure 3A–H; LSID urn:lsid:zoobank.org:act:1BC155B9-8599-44EF-919F-975620B45DCF)

**Etymology.** The specific name honors R. J. Tillyard, a distinguished pioneer in the study of fossil Mecoptera.

**Type locality and horizon.** Yinping Mountain, Chaohu City, Anhui Province, China; Yinping Formation (Capitanian).

**Diagnosis.** Forewings densely ornamented with rounded and oval-colored spots; costal area greatly expanded at apex of Sc_1_; Rs_1+2_ twice as long as Rs_3+4_. Hindwings with short, single-branched Sc; Rs and M both four-branched.

**Material.** Holotype, NIGP205288a–c (Figure 1A–C), specimen broken into three pieces. NIGP205288a (Figure 1A) preserves two hindwings and one forewing attached into the body; another isolated forewing preserved at 2 cm away. NIGP205288b (Figure 1B) preserves the body and two hindwings. NIGP205288c (Figure 1C) preserves a single forewing, representing the counterpart of the isolated forewing in NIGP205288a.

Paratypes, NIGP205289, NIGP205290a, b (part and counterpart), NIGP205291a, b, NIGP205292a, b, NIGP205293a, b, and NIGP205294a, b, each representing a complete forewing; NIGP205295 a, b (part and counterpart), a hindwing missing a small part of wing base; NIGP205296, a forewing lacking apex, middle part distorted and missing; NIGP205297 and NIGP208853, incomplete forewings.

**Description.** Holotype, NIGP205288, female; thorax and head structures poorly preserved. I–VII abdominal segments with clear boundary; I–IV subequal in length, width slowly tapering posteriorly; VI and VII equal in length, half as long as segment V; segment II–IV each with a sclerotized drum-shaped tergite, the other segments indistinct; one hind leg preserved, trochanter drum-shaped; femur robust, 1.7 mm long, 0.2 mm wide; tibia much thinner than femur, partially preserved.

Forewings elongate, 6.3 mm long, 2.0 mm wide (L/W = 3.2) (Figure 3B), apex narrowly convex, covered with dense rounded and oval pigmented spots, interspersed with larger patches, colored markings denser along wing margin and posterior basal region; Sc initially parallel to costa, distally forked into two branches, costal area expanded along Sc_1_, a short crossvein connecting Sc_1_ and R_1_ at narrowest area; Sc_2_ and Sc_3_ crossvein-like, origin of Sc_2_ at level of Rs fork, and Sc_3_ at level of origin of Rs; R_1_ single-branched, curving down after entering pigmented, elongated pterostigma; pterostigma fused with adjacent colored markings; Rs with four branches, Rs_1+2_ 2.1 × length of Rs_3+4_, two crossveins connecting Rs and M; M with six branches, M_1+2_ 4 × length of M_3+4_, M_4_ longer than its branches, one crossvein between M_1_ and M_2_, another between M_1+2_ and M_3_; crossvein m-cua connecting fork of M_3+4_ and CuA; Rs fork slightly distal to M fork; CuA single, curved apically, M_5_ very short, or M, M_5_, CuA, and CuA base meeting at one point, forming a “X”; CuP single, smoothly curved at apex; crossvein cua-cup transverse in one wing, obliquely inclined in the other wing; three anal veins detected: A_1_ straight, A_2_ undulate, A_3_ short and straight; one crossvein connecting bases of A_1_ and A_2_.

Hindwings smaller and rounder than forewings, devoid of colored markings, 4.4 mm long, 1.7 mm wide (L/W = 2.4); Sc single and short, terminating at the same level as Rs fork; R_1_ distally forked into two branches, covered by pigmented lentoid pterostigma; Rs with four branches; one crossvein between Rs_2_ and Rs_3_; Rs_1+2_ 1.6 × length of Rs_3+4_; M with four branches; M_1+2_ 2 × length of M_3+4_; one crossvein between M_1+2_ and end of Rs_3+4_; crossvein m-cua connecting mid M_4_ and CuA; Rs fork proximal to M fork; CuA and CuP single; CuA fused with M stem for long distance; one anal vein present; other details not preserved.

Paratypes, NIGP205289 (Figure 2A and Figure 3D), 5.5 mm long, 2.0 mm wide (L/W = 2.8), humeral vein present; Sc_3_ present; Rs_1+2_ 2.2 × length of Rs_3+4_; M_1+2_ 5.3 × length of M_3+4_, M_4_ shorter than M_4a_ and M_4b_; one crossvein between Rs_2_ and Rs_3_, and between Rs_3_ and Rs_4_, two crossveins between Rs and M, crossvein m-cua connecting basal M_4_ and CuA; NIGP205290 (Figure 2B and Figure 3E), 5.5 mm long, 2.0 mm wide (L/W = 2.8), Sc_3_ undetected, crossvein sc_1_-r_1_ moderately long, Rs_1+2_ 2.2 × length of Rs_3+4_, M_1+2_ 5.0 × length of M_3+4_, three crossveins connecting Rs and M, crossvein m-cua connecting basal M_4_ and CuA; NIGP205291 (Figure 3F), 6.4 mm long, 2.0 mm wide (L/W = 3.2), wing relatively elongated, Sc_2_ relatively long, Sc_3_ undetected, Rs_1+2_ 2.6 × length of Rs_3+4_, M_1+2_ 5.4 × length of M_3+4_, M_2_ relatively short, M_4_ shorter than M_4a_, one crossvein connecting Rs_3_ and Rs_4_, crossvein m-cua connecting basal M_4_ and CuA; NIGP205292 (Figure 3G), 5.4 mm long, 1.9 mm wide (L/W = 2.8), Sc_2_ relatively short, Sc_3_ at level of Rs origin, Rs_1+2_ 2.5 × length of Rs_3+4_, M_1+2_ 5.6 × length of M_3+4_, M_4_ stem longer than M_4a_ and M_4b_, sc_1_-r_1_ moderately short, one crossvein between Rs_1_ and Rs_2_, and mid Rs_3_ and Rs_4_, m-cua connected basal M_4_ and CuA; NIGP205293 (Figure 3H), 4.8 mm long, 2.0 mm wide (L/W = 2.8), Sc_1_ moderately short, Sc_2_ very short, Sc_3_ undetected, costal area narrow, Rs_1+2_ 1.7 × length of Rs_3+4_, M_1+2_ 3.0 × length of M_3+4_, M_2_ fork rounded; NIGP205294, 5.0 mm long (as preserved), 1.8 mm wide, Sc moderately short, Sc_3_ undetected, Rs_1+2_ 2.1 × length of Rs_3+4_, M_1+2_ 5.0 × length of M_3+4_, M_4_ longer than M_4a_ and M_4b_, sc_1_-r_1_ short, one crossvein connecting Rs_3_ and Rs_4_, two crossveins connecting Rs and M, m-cua connecting basal M_4_ and CuA; NIGP205295 (Figure 3C), 4.2 mm long, 1.6 mm wide, hindwing, R_1a_ forming the boundary of pterostigma basal; Rs_1+2_ 2.0 × length of Rs_3+4_, M_1+2_ 2.7 × length of M_3+4_, two straight anal veins present; NIGP205296, 4.9 mm long (as preserved), 1.7 mm wide, wing deformed in middle part, Sc_3_ proximal to origin of Rs.

**Figure 2 insects-16-01130-f002:**
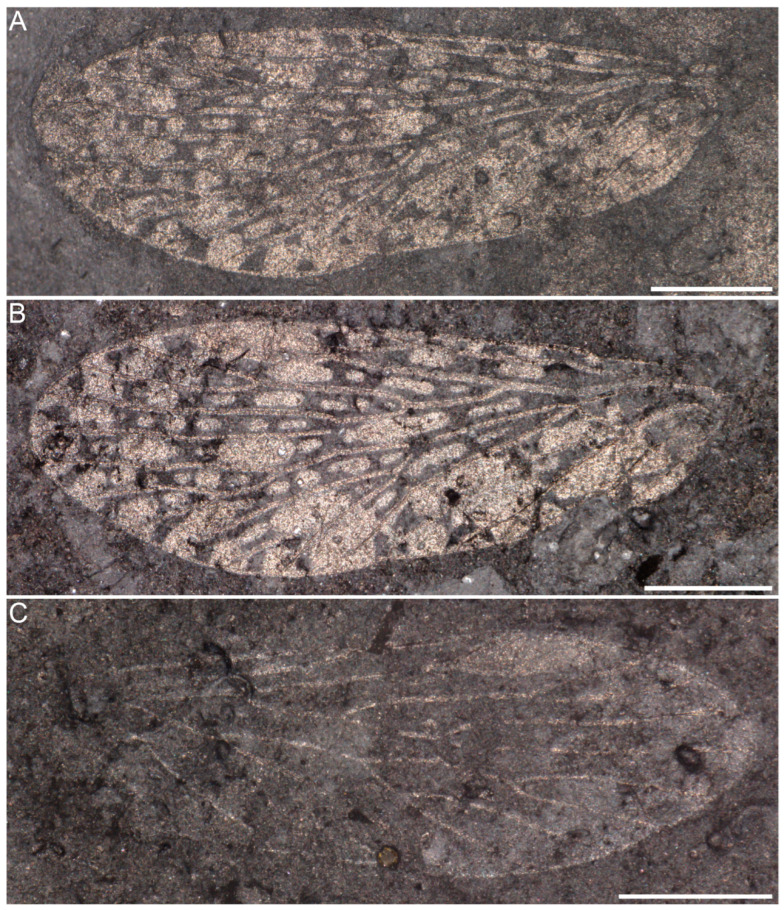
Paratypes of *Mesochorista tillyardi* Lian and Huang, sp. nov. (**A**) NIGP205289. (**B**) NIGP205290a. (**C**) NIGP205295a. Photographs were captured under vertical reflected light after being moistened with 75% alcohol. Scale bars = 1 mm in (**A**–**C**).

Other fragmentary specimens include NIGP205297 and NIGP208853.

**Figure 3 insects-16-01130-f003:**
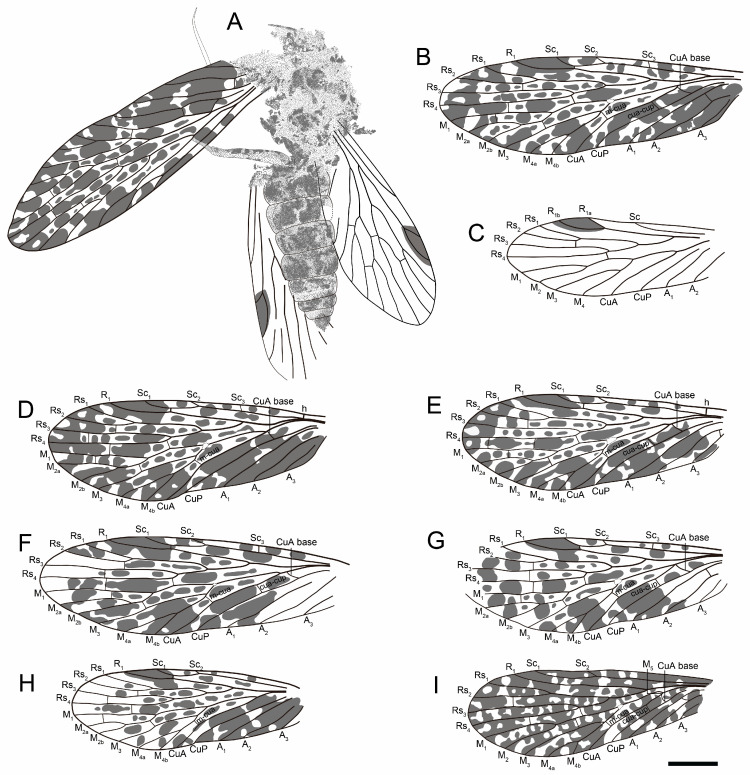
Line drawings. (**A**–**H**) *Mesochorista tillyardi* Lian and Huang, sp. nov. (**A**) General habitus, NIGP205288a (holotype). (**B**) NIGP205288c (holotype), forewing. (**C**) NIGP205295, hindwing. (**D**) NIGP205289, forewing. (**E**) NIGP205290, forewing. (**F**) NIGP205291, forewing. (**G**) NIGP205292, forewing. (**H**) NIGP205293, forewing. (**I**) *Mesochorista yinpingensis* Lian and Huang, sp. nov., NIGP205298, forewing. Scale bar = 1 mm in (**A**–**I**).

**Remarks.** *Mesochorista tillyardi* Lian and Huang, sp. nov. represents the dominant *Mesochorista* species in Chaohu City, comprising 11 specimens that exhibit a stable venational pattern, with only minor variations present in the arrangement of crossveins and the relative position of some forks (such as Sc, Rs_1+2_ and Rs_3+4_, M_1+2_ and M_3+4_). Sc_3_ is a very faint vein in this species, and its absence in some line drawings is due to non-observation, possibly as a result of preservation. *Mesochorista tillyardi* Lian and Huang, sp. nov. can be confidently placed to Permochoristinae by the presence of two to three Sc branches (one or two anterior veinlets), a four-branched Rs with Rs_1+2_ much longer than Rs_3+4_, and a six-branched M in which both M_2_ and M_4_ are bifurcated [22]. They can further be placed within the genus *Mesochorista* due to the costal space being narrow, and the presence of one or two crossvein-like Sc anterior veinlets (Sc_2_ and Sc_3_).

*Mesochorista tillyardi* Lian and Huang, sp. nov. is characterized by its forewings covered with dense pigmented spots interspersed with several colored patches, and by Sc_1_ closely approximating R_1_. This combination of characters readily differentiates it from other related species. It can be further distinguished from *M*. *proavita* (the type species of *Mesochorista*) by its smaller wing size (4.7–6.4 mm vs. preserved 10.5 mm), Rs_1+2_ not strongly curved upwards, and a lower Rs_1+2_/Rs_3+4_ ratio (1.7–2.6 vs. 3.6) [23,24]; from *M*. *conjunctiva* by its Rs_1+2_ not strongly curved upwards, and a lower Rs_1+2_/Rs_3+4_ ratio (1.7–2.6 vs. 3.8) [25]; from *M*. *australica* by its smaller wing size (4.7–6.4 mm vs. ca. 12 mm (preserved 8.0 mm)), M_4_ fork sharp instead of rounded, and a higher M_1+2_/M_3+4_ ratio (4.6–5.4 vs. 2.9) [8]; from *M*. *affinis* by lacking strongly sinuate CuA apex [7]; from *M*. *sinuata* (Handlirsch, 1939) from the Lower Jurassic of Dobbertin, which represents the latest occurrence of *Mesochorista*, by its apical costal area expanded, R_1_ curved near apex instead of sinuous, M_2_ forked instead of single, and a long stem of M_4_; from *M*. *angustipennis* by its smaller and broader wings (4.7–6.4 mm vs. at least 12.0 mm; L/W ratio = 2.8–3.2 vs. 3.9), and Sc_1+2_ fork at same level as Rs fork instead of strongly proximal to Rs fork [6].

The hindwing somewhat resembles that of *Mesochorista prospera*, but its Sc is much shorter, and M with four branches instead of five branches [26]. It differs from the hindwings of *M*. *wolbromae* and *M*. *sokolovensis* by the shorter Sc, and broader costa space [26].


** **


*Mesochorista yinpingensis* Lian and Huang, sp. nov.

(Figure 3I and Figure 4; SID urn:lsid:zoobank.org:act:2D63D30A-AB5B-4F62-A23B-2FFCABAE3A01)

**Etymology.** The species epithet is derived from Yinping Mountain, where the fossil was collected.

**Type locality and horizon.** Yinping Mountain, Chaohu City, Anhui Province, China; Yinping Formation (Capitanian).

**Diagnosis.** Forewing with dense, irregular pigmented patches; Sc_1_ relatively long, distal to Rs_1+2_ fork; M with five branches.

**Material.** Holotype, NIGP205298a, b, with part and counterpart. Two forewings, one hindwing overlapped with a forewing, and some blurry body structures are preserved; the venation of hindwing is partially identified.

**Description.** Forewing elongate and rounded, widest at wing 3/4 length, tapering basally, wing base narrow; forewing 5.0 mm long, 1.8 mm wide (L/W = 1.9); covered with dense irregular pigmented patches, pigmentation dominant over hyaline area; pterostigma pigmented, boundary unclear, overlying apical Sc; Sc parallel to costa, forked into two branches at mid wing, costal area along Sc_1_ not expanded, Sc_2_ short and oblique, Sc fork distal to Rs fork; Rs with four branches, Rs_1+2_ 1.9 × length of Rs_3+4_; M with five branches, M_1+2_ 2.8 × length of M_3+4_, M_4_ fork small, M_4_ slightly longer than M_3+4_, M fork unsclerotized; Rs fork distinctly distal to M fork; CuA and CuP single, CuA base crossvein-like, M_5_ short, CuA base and crossvein cua-cup transverse; three anal veins present, one oblique crossvein connecting the base of A_1_ and A_2_; one crossvein connecting Rs_2_ and Rs_3_, M_2_ and M_3_, and A_1_ and A_2_, respectively, two crossveins connecting Rs and M.

**Remarks.** The species *M. yinpingensis* Lian and Huang, sp. nov. can be assigned to the subfamily Permochoristinae based on the presence of one anterior veinlet on Sc, a four-branched Rs with Rs_1+2_ significantly longer than Rs_3+4_*,* and a five-branched M with a forked M_4_.

*Mesochorista yinpingensis* sp. nov. differs from the co-occurred *M*. *tillyardi* Lian and Huang, sp. nov. by having its forewing widest closer to the apex, colored markings denser and in an irregular shape, Sc_1_ longer, costal area along Sc_1_ not expanded, lacking the crossvein between Sc_1_ and R_1_, and M with five branches instead of six. The presence of a five-branched M allows the new specimens to be readily distinguished from other related species. Among species with a five-branched M, the new species can be further differentiated as follows: it differs from the holotype of *M*. *asiatica* (Martynova, 1948) by its Sc fork distal to (rather than proximal to) Rs fork, R_1_ apex smooth (vs. distinctly curved), Rs_1+2_/Rs_3+4_ ratio smaller (1.9 vs. 3.1), and Rs fork distinctly distal to M fork (vs. nearly aligned) [17,27]; from *Yanorthophlebia hebeiensis* Ren, 1995 (Mecoptera: incertae familiae) by having a smaller wing size (length 5.2 mm vs. 7.2 mm), Rs fork distinctly distal to M fork (vs. nearly at the same level), M_4_ fork smaller, and CuA base transverse instead of horizontal [28,29].

The new species resembles species of *Permeca* Novokshonov, 1995, except for the arrangement of Sc_1_. Besides, it differs from *Permeca tatarica* by its colored markings denser and not arranged in bands, Rs_1+2_ fork deeper, M_2_ single instead of with one small fork, M_4_ fork small, and apical CuA rather straight instead of sinuous [30]; differs from *Permeca chaohuensis* Lian, Cai and Huang, 2022 from the same beds by its wing larger (5.0 mm long vs. less than 4.0 mm), the absence of a strong crossvein between Sc_1_ and R_1_, M_4_ fork small, apical CuA rather straight instead of sinuous, and CuA base transverse instead of horizontal [19].

## 4. Discussion

The family Permochoristidae is a “wastebasket” taxon that includes the majority of Permian mecopterans, which exhibit highly diverse venational patterns. Over the past century, approximately 40 genera and around 200 species have, at various times, been assigned to this family [22,31,32]. A systematic revision of this megadiverse family would constitute an extensive undertaking far beyond the scope of this paper. The Family Permochoristidae was established based on two poorly preserved specimens from the Upper Permian Belmont insect bed, and Tillyard [8] designated specimen No. 24, lacking the costal area, as the holotype of *Permochorista australica*, the type species of the type genus of the family. Meanwhile, he described another species, *Permochorista mitchelli*, based on specimen No. 26, which consists of a preserved basal portion of a wing only, and he illustrated it with a line drawing showing the Sc vein with multiple veinlets, which may be incorrect. Sc with multiple branches is a primitive character which is not a character of *Permochorista* [22]. Moreover, some taxa previously illustrated by Tillyard as possessing multiple Sc branches have been proven otherwise upon reexamination. For example, reexaminations of *Mesochorista proavita* Tillyard, 1916 and *Mesochorista anglicana* (Tillyard 1933) revealed that neither species exhibits multiple veins along the costal and R_1_ regions (Willmann [24]: Figure 2). Tillyard [7] described two additional, well-preserved species from the same locality, both showing only one short anterior veinlet (Sc_2_) in the costal area. Later, many species with a similarly short anterior Sc veinlet were assigned to this genus [6].

*Mesochorista* was established based on a holotype specimen collected from the Middle Triassic of Denmark Hill, Queensland, Australia [23]. Initially, this species was misinterpreted as having three additional crossvein-like structures between Sc_2_ and h. Willmann [24] corrected this interpretation, clarifying that there is only one vein (Sc_3_) between Sc_2_ and h, and provided further details on the crossveins in other regions of the wing. *Mesochorista* shares venational similarity with *Permochorista*, and they have been synonymized by Riek [31]. However, Novokshonov [22] argued that this synonymization is not sufficiently justified, based on his observation that one of the anterior crossveins (interpreted as the distal branch of Sc, possibly Sc_2_) is inclined to a certain degree, while the other (the proximal veinlet, possibly Sc_3_) appears nearly desclerotized or absent. Nonetheless, this distinction is insufficient to serve as a reliable diagnostic feature above the species level and is of limited practical utility in taxonomic application. Therefore, we follow the opinion of Riek and treat *Permochorista* as a junior synonym of *Mesochorista*. According to personal communication with Alexey Bashkuev, he also holds this view.

It is worth noting that Riek [31] considered *Mesochorista* to possess only two Sc branches (one anterior Sc veinlet). However, according to the reconstructed venation by Willmann [24], the type species of *Mesochorista* exhibits three Sc branches (two anterior Sc veinlets). Based on our examination of *Mesochorista* specimens from the Middle Triassic Tongchuan and Guadalupian Permian Yinping entomofaunas, Sc_3_ are often extremely faint and may not be discernible due to poor preservation [33]. This suggests that it may have been overlooked in previously described species. Consequently, the presence or absence of Sc_3_ is not regarded as a definitive diagnostic character of *Mesochorista*.

Previously, only two species of *Mesochorista* have been reported from China, both from the Middle Triassic Tongchuan entomofauna: *M. conjunctiva* (Guo and Hong, 2003), and *M. tongchuanensis* Lian, Cai and Huang, 2022. *M. tillyardi* Lian and Huang, sp. nov. share some venational similarities with *M. tongchuanensis* and *M. conjunctiva*, including a narrow costal area, an expanded Sc_1_ approaching R_1_, and a typically faint Sc_3_, indicating a close phylogenetic relationship. It is worth noting that the new species from the Yinping entomofauna exhibit denser colored markings which are lacking in those from Tonchuan entomofauna, indicating that the colored pattern may be regionally distinctive.

These newly discovered fossils represent the first occurrence of *Mesochorista* with preserved body structures, thereby broadening our knowledge of its morphology. In addition, new discoveries extend the known geographical distribution of *Mesochorista* and enhance our understanding of its paleobiodiversity.

## Figures and Tables

**Figure 1 insects-16-01130-f001:**
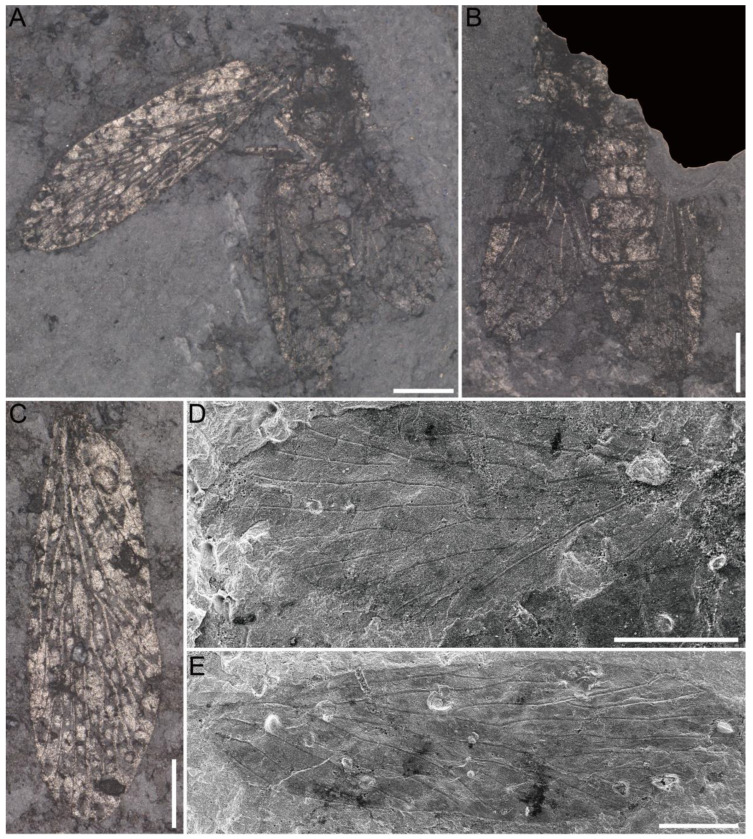
*Mesochorista tillyardi* Lian and Huang, sp. nov., NIGP205288 (holotype). (**A**) NIGP205288a, part, general habitus. (**B**) NIGP205288b, counterpart. (**C**) NIGP205288c, a forewing, the other counterpart of NIGP205288a, preserved 2 cm away from body. (**D**) Enlargement of hindwing from (**B**), scanning electron microscope image. (**E**) Enlargement of forewing from (**A**), scanning electron microscope image. (**A**–**C**) were captured under vertical reflected light after being moistened with 75% alcohol. Scale bars = 1 mm in (**A**–**E**).

**Figure 4 insects-16-01130-f004:**
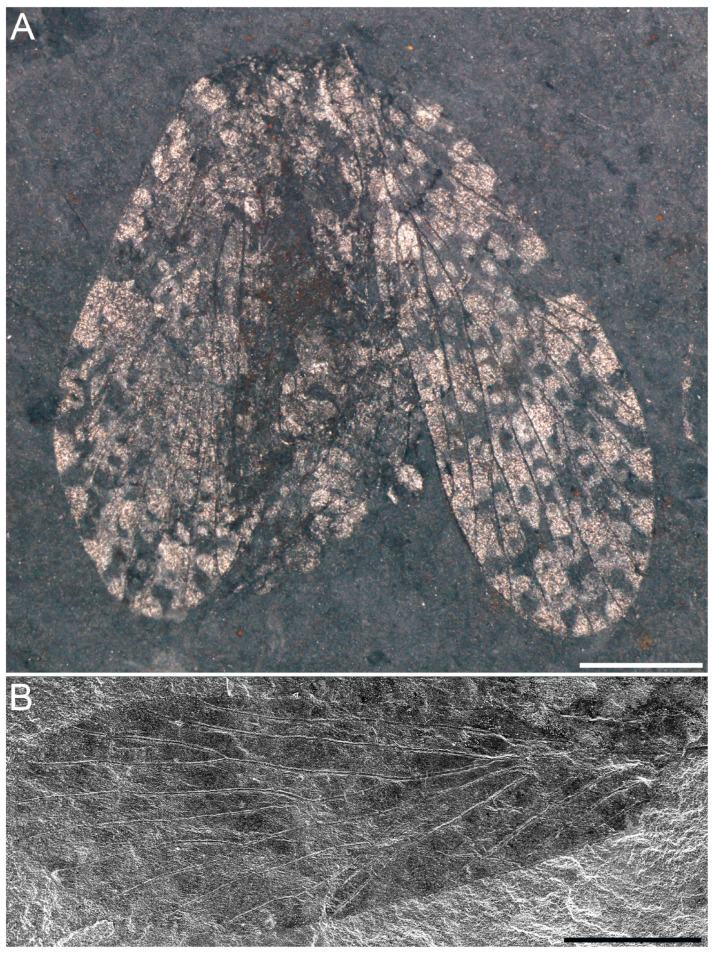
*Mesochorista yinpingensis* Lian and Huang, sp. nov., holotype (NIGP205298). (**A**) NIGP205298a. (**B**) Enlargement of forewing from (**A**), scanning electron microscope image. Scale bars = 1 mm in (**A**,**B**).

## Data Availability

All data generated during this study are included in this published article. All the specimens are housed in the Nanjing Institute of Geology and Palaeontology, Chinese Academy of Sciences, Nanjing, China.

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
