# Peer review of "Two New Species of Mesochorista (Insecta, Mecoptera, Permochoristidae) from the Guadalupian Yinping Formation of Chaohu, Eastern China†"

_insects, 2025, doi:10.3390/insects16111130_

Round 1

Reviewer 1 Report

Comments and Suggestions for Authors

See the attached document.

Comments on the Quality of English Language

English of the manuscript is generally good, I found only a few minor grammatical errors that need correction.

Author Response

Comments 1: It's over 800 now. Dozens of new species were described since the Color Atlas (2022) was published.

Response 1: Yes, thank you for pointing this out. I agree with the comment, and have changed the number from 700 to 800. I have also added a new reference concerning the total number of known species of Mecoptera.

Comments 2: North America or North and South America (but delete Brazil then)

Response 2: I agree with the comment. I have changed the America to North America.    

Comments 3: please add the authorhips names and dates for all taxa when first mentioned in text (especially for the species of Mesochorista, such as M. sinuata, M. affinis, etc.).

Response 3: Thank you for pointing this out, I have added accordingly.

Comments 4: suggest to specify ("up to 10 mm" or so)

Response 4: Thank you for pointing this out. I have changed it to “Forewings ca. 6–11 mm long”

Comments 5: In Mesosoic species five-branched media is quite common.

Response 5: Thank you for pointing this out. I have revised it to “sometimes”, which is more reasonal.

Comments 6: a, b ?

Response 6: Thank you for pointing this out. NIGP205295 has part and counterpart, I have added “a, b (part and counterpart)”.

Comments 7: Note that this species is Jurassic (the latest occurrence of the genus) and it has 5-branched M.

Response 7: Thank you for pointing this out, I have added accordingly.

Comments 8: I would suggest comparing M. yinpingensis not only with other species of Mesochorista, but also with the species of Permeca, especially Permeca tatarica Novokshonov, 1995, which looks very similar in appearance (although its Sc is quite different from Mesochorista).

Response 8: Thank you for pointing this out, I have added accordingly.

Comments 9:  “Mâ‚„. A single Mâ‚‚ vein is relatively common within Permochoristidae and may represent an intraspecific abbreviation” This sentence is confusing and should be rephrased. I guess you mean that M. yinpingensis might represent an intraspecific variation of M. tillyardi, is it? However, it is not supported by the cited papers: they say that in M. asiatica, five-branched M is common, while short forks on M2 are rare variations observed in only a few specimens. Your case is quite the opposite.

Response 9: Thank you for pointing this out. You are right. I only intended to emphasize that Mesochorista yinpingensis can be ruled out as an intraspecific variant of Mesochorista tillyardi. However, as you mentioned, my case is incorrect, and we have decided to remove this sentence.

Comments 10: Technically, Yanorthophlebia hebeiensis Ren, 1995 was never attributed to Mesochorista.

Novokshonov (1997) transferred it to Liassochorista, which he believed to be a different genus from Mesochorista (contrary to Ansorge, 1995, who synonymized Liassochorista under Mesochorista). Therefore, M. hebeiensis (Ren 1995) must be a new combination here.

However, according to Bashkuev (in Kopylov et al., 2020), Y. hebeiensis should not be considered as Permochoristidae, but rather belongs to crown-group scorpionflies, Panorpoidea, based on numerous new specimens related to Yanorthophlebia discovered in the Lower Cretaceous deposits. Unfortunately, the type specimen of Y. hebeiensis is not available for re-examination, thus the question is still open. With Y. hebeiensis discarded, the latest occurences of Permochoristidae would be in Lower and lower Middle Jurassic.

Response 10: Thank you for pointing this out. I missed the information that Bashkuev discovered some new specimens of Yanorthophlebia with body structures. Since Yanorthophlebia is a valid taxon, I have changed its generic assignment from Mesochorista to Yanorthophlebia.

Reviewer 2 Report

Comments and Suggestions for Authors

This manuscript insects-3932824, entitiled “New species of Mesochorista (Insecta, Mecoptera, Permochoristidae) from the Guadalupian Yinping Formation of Chaohu City, China”, describes two new species of Mesochorista in the extinct family Permochoristidae in Mecoptera. Generally speaking, this manuscript is well written, and the pictures are of good quality. Therefore, I recommend it to be published in Insects after a minor revision.

  1. Title: to be more precise, I recommend the authors to add “Two” at the beginning of the title as “Two new species of Mesochorista…”ï¼›
  2. Simple Summary: Line 15, “crow group”, I wonder if it is a misspelling of “crown group”. L16, “fossil records from China remains scarce”, the subject “records” is plural, and is not consistent with the verb “remains”ï¼›
  3. L19-20: The statement “we propose that Permochorista should be considered a junior synonym of Mesochorista” could be misunderstood as the original opinion of the authors. In fact, when I go through the manuscript, it is Riek (1953) who first treated Permochorsta as a junior synonym of Mesochorista. Please modify this properly;
  4. Introduction: L50, the family Permochoristidae should not be printed in italics. Permochoristidae Tillyard, 1916 is not consistent with Permochoristidae Tillyard, 1917 in the context and the referenceï¼›
  5. L85-86: Please reverse the subfamily and family. In other words, please put the Family before the Subfamily name;
  6. L235: “The new species” is redundant here, and can be deletedï¼›
  7. Discussion: A family is established on a type genus, not specimens. Hensce, in L259, “the type specimen of this family” is not a proper expression, since the type of a family is a genus, not a type specimen.
  8. Similarly, L287, “the holotype of Mesochorista” are also a wrong expression, because the type of a genus is a species, not a specimen. Please modify properly。
Comments on the Quality of English Language

This manuscript insects-3932824, entitiled “New species of Mesochorista (Insecta, Mecoptera, Permochoristidae) from the Guadalupian Yinping Formation of Chaohu City, China”, describes two new species of Mesochorista in the extinct family Permochoristidae in Mecoptera. Generally speaking, this manuscript is well written, and the pictures are of good quality. Therefore, I recommend it to be published in Insects after a minor revision.

  1. Title: to be more precise, I recommend the authors to add “Two” at the beginning of the title as “Two new species of Mesochorista…”ï¼›
  2. Simple Summary: Line 15, “crow group”, I wonder if it is a misspelling of “crown group”. L16, “fossil records from China remains scarce”, the subject “records” is plural, and is not consistent with the verb “remains”ï¼›
  3. L19-20: The statement “we propose that Permochorista should be considered a junior synonym of Mesochorista” could be misunderstood as the original opinion of the authors. In fact, when I go through the manuscript, it is Riek (1953) who first treated Permochorsta as a junior synonym of Mesochorista. Please modify this properly;
  4. Introduction: L50, the family Permochoristidae should not be printed in italics. Permochoristidae Tillyard, 1916 is not consistent with Permochoristidae Tillyard, 1917 in the context and the referenceï¼›
  5. L85-86: Please reverse the subfamily and family. In other words, please put the Family before the Subfamily name;
  6. L235: “The new species” is redundant here, and can be deletedï¼›
  7. Discussion: A family is established on a type genus, not specimens. Hensce, in L259, “the type specimen of this family” is not a proper expression, since the type of a family is a genus, not a type specimen.
  8. Similarly, L287, “the holotype of Mesochorista” are also a wrong expression, because the type of a genus is a species, not a specimen. Please modify properly.

Author Response

Comments and Suggestions for Authors

This manuscript insects-3932824, entitiled “New species of Mesochorista (Insecta, Mecoptera, Permochoristidae) from the Guadalupian Yinping Formation of Chaohu City, China”, describes two new species of Mesochorista in the extinct family Permochoristidae in Mecoptera. Generally speaking, this manuscript is well written, and the pictures are of good quality. Therefore, I recommend it to be published in Insects after a minor revision.

  1. Title: to be more precise, I recommend the authors to add “Two” at the beginning of the title as “Two new species of Mesochorista…”ï¼›

Thank you for your comments, I have revised the title to “Two new species of Mesochorista (Insecta, Mecoptera, Permochoristidae) from the Guadalupian Yinping Formation of Chaohu City, China”

  1. Simple Summary: Line 15, “crow group”, I wonder if it is a misspelling of “crown group”. L16, “fossil records from China remains scarce”, the subject “records” is plural, and is not consistent with the verb “remains”ï¼›

Thank you for your comments. I have revised accordingly.

  1. L19-20: The statement “we propose that Permochorista should be considered a junior synonym of Mesochorista” could be misunderstood as the original opinion of the authors. In fact, when I go through the manuscript, it is Riek (1953) who first treated Permochorsta as a junior synonym of Mesochorista. Please modify this properly;

Thank you for your comments. Yes, my expression was incorrect. I have revised “propose” to “support.”

  1. Introduction: L50, the family Permochoristidae should not be printed in italics. Permochoristidae Tillyard, 1916 is not consistent with Permochoristidae Tillyard, 1917 in the context and the referenceï¼›

Thank you for your comments. I have revised accordingly.

  1. L85-86: Please reverse the subfamily and family. In other words, please put the Family before the Subfamily name;

Thank you for your comments. I have revised accordingly.

  1. L235: “The new species” is redundant here, and can be deletedï¼›

Thank you for your comments. I have revised accordingly.

  1. Discussion: A family is established on a type genus, not specimens. Hensce, in L259, “the type specimen of this family” is not a proper expression, since the type of a family is a genus, not a type specimen.

Thank you for pointing this out. I have  revised this sentence as “The Family Permochoristidae was established based on two poorly preserved specimens from the Upper Permian Belmont insect bed, and Tillyard[8] designated the specimen No. 24, lacking the costal area, as the holotype of Permochorista australica,the type species of the type genus of the family”

  1. Similarly, L287, “the holotype of Mesochorista” are also a wrong expression, because the type of a genus is a species, not a specimen. Please modify properly。

Thank you for your comments. I have revised “the holotype of Mesochorista” to “the type species of Mesochorista

Reviewer 3 Report

Comments and Suggestions for Authors

Dear colleagues,

it was a pleasure for me to review your manuscript. You described two new species in a very accurate and detailed manner. One of the species is represented by several specimens, where you discussed the intraspecific differences and explained taphonomic influences. In your discussion you highlighted the current problematic taxonomic situation within Permochoristidae and provided a useful discussion about the relationships between species currently recognized as belonging to Permochorista and Mesochorista. The figures are well prepared and share a consistent design, the drawings proved very helpful (more venational labels could further improve them).

There are two points, where I think the manuscript could still be improved:

1) I think it would be fruitful to highlight which diagnostic characters of Mesochorista can be interpreted as derived and which are plesiomorphic. This could, depending on the outcome, either strengthen or weaken Mesochorista as a taxonomic unit. In both cases, this will be helpful for future studies.

2) I am not familiar with the Yinping fossil site, but from what I could see from photographs of insects from this site, I would recommend to be more cautious with the interpretation of “colour patterns”. Usually color patterns in insect wings are not so much constrained by the underlying wing cells, but can form irregular patterns. In your fossils, the colour pattern is very much constrained to the areas between veins. In another publication of you (Lian et al. 2023), I found specimens that have the metallic reflective surface spanning the entire wing. This leads me to the hypothesis that the shiny reflective surface is actually just derived from the taphonomically altered chitin and the darker areas are areas, where the organic remnants are peeled off from the rock or covered by a very thin layer of sediment. By a close inspection of the specimens, it should be possible to test this alternative hypothesis. This could potentially also be visible under the SEM at higher magnification. I do not think that the outcome would drastically change your systematic interpretation but knowing more about this potential taphonomic artefact, could prevent future taxonomic mistakes (or strengthen decisions in case you can support the interpretation of the patches as biological features).

Please also review my suggestions and corrections in the annotated PDF.

Congratulations on this contribution and best wishes,

Lian, X.; Cai, C.; Huang, D. Sinoagetopanorpidae fam. nov., a New Family of Scorpionflies (Insecta, Mecoptera) from the Guadalupian of South China. Insects 2023, 14, 96. https://doi.org/10.3390/insects14010096

Author Response

There are two points, where I think the manuscript could still be improved:

  • I think it would be fruitful to highlight which diagnostic characters of Mesochorista can be interpreted as derived and which are plesiomorphic. This could, depending on the outcome, either strengthen or weaken Mesochoristaas a taxonomic unit. In both cases, this will be helpful for future studies.

Thank you for pointing this out. An additional Remarks section discussing this aspect has been included following the Revised Diagnosis of Mesochorista.

  • I am not familiar with the Yinping fossil site, but from what I could see from photographs of insects from this site, I would recommend to be more cautious with the interpretation of “colour patterns”. Usually color patterns in insect wings are not so much constrained by the underlying wing cells, but can form irregular patterns. In your fossils, the colour pattern is very much constrained to the areas between veins. In another publication of you (Lian et al. 2023), I found specimens that have the metallic reflective surface spanning the entire wing. This leads me to the hypothesis that the shiny reflective surface is actually just derived from the taphonomically altered chitin and the darker areas are areas, where the organic remnants are peeled off from the rock or covered by a very thin layer of sediment. By a close inspection of the specimens, it should be possible to test this alternative hypothesis. This could potentially also be visible under the SEM at higher magnification. I do not think that the outcome would drastically change your systematic interpretation but knowing more about this potential taphonomic artefact, could prevent future taxonomic mistakes (or strengthen decisions in case you can support the interpretation of the patches as biological features).

Thank you for pointing this out.

It is quite clear that the colored markings are intrinsic to the wings rather than taphonomic artefacts. On one hand, Mesochorista tillyardi Lian and Huang, sp. nov. is established based on twelve forewing specimens (including two forewings in the holotype), all of which share a similar color pattern, with certain colored spots confined to the areas between veins. If these markings were taphonomic artefacts, it would be impossible for eleven wings to exhibit the same pattern. On the other hand, we examined some specimens using SEM. The SEM images also show similar color patterns, and do not reveal any evidence of areas where organic remnants were peeled off from the rock surface or covered by a thin sedimentary layer

Images of SEM and microscope (NIGP205290a) (see the World file)

Please also review my suggestions and corrections in the annotated PDF.

Response to annotated PDF.

Comments 1: how can you tell this (M fork unsclerotized) from one specimen? Could it not be just a poor preservation in this part of the wing?

Reseponse: M fork unsclerotized is quite common in Mecoptera, such as Permochoristidae, Orthophlebiidae, and the extant Panorpidae.

The other comments have been addressed accordingly.
